# Effect of environmental DNA sampling resolution in detecting nearshore fish biodiversity compared to capture surveys

Ben Millard-Martin[1], Kate Sheridan[1], Evan Morien[2], Matthew A. Lemay[2], Margot Hessing-Lewis[2], Rute B.G. Clemente-Carvalho[2] and Jennifer M. Sunday[1]

[1] Department of Biology, McGill University, Montreal, Quebec, Canada
[2] Department of Biodiversity, Hakai Institute, Victoria, British Columbia, Canada

Corresponding author
Ben Millard-Martin,
benjamin.millard-martin@mail.mcgill.ca

## ABSTRACT

Sampling and sequencing marine environmental DNA (eDNA) provides a tool that can increase our ability to monitor biodiversity, but movement and mixing of eDNA after release from organisms before collection could affect our inference of species distributions. To assess how conditions at differing spatial scales influence the inferred species richness and compositional turnover, we conducted a paired eDNA metabarcoding and capture (beach seining) survey of fishes on the coast of British Columbia. We found more taxa were typically detected using eDNA compared to beach seining. eDNA identified more taxa with alternative habitat preferences, and this richness difference was greater in areas of high seawater movement, suggesting eDNA has a larger spatial grain influenced by water motion. By contrast, we found that eDNA consistently missed low biomass species present in seining surveys. Spatial turnover of communities surveyed using beach seining differed from that of the eDNA and was better explained by factors that vary at small (10–1000s meters) spatial scales. Specifically, vegetation cover and shoreline exposure explained most species turnover from seining, while eDNA turnover was not explained by those factors and showed a distance decay pattern (a change from 10% to 25% similarity from 2 km to 10 km of distance), suggesting unmeasured environmental variation at larger scales drives its turnover. Our findings indicate that the eDNA sample grain is larger than that of capture surveys. Whereas seining can detect differences in fish distributions at scales of 10s–100s of meters, eDNA can best summarize fish biodiversity at larger scales possibly more relevant to regional biodiversity assessments.

## INTRODUCTION

Coastal marine environments provide critical nursery and foraging habitats for a variety of commercially, culturally, and ecologically important fish species (*Suchanek, 1994*; *Duffy, 2006*; *Sheaves et al., 2014*; *Olson et al., 2019*). These nearshore habitats are also highly affected by human activity and invasive species (*Lotze et al., 2006*; *Kelly et al., 2016*), impacting the species they support. Efficiently monitoring species distributions in nearshore habitats can help us understand the effects of these disturbances, how they vary

in space and time, and which conservation and management efforts are most effective in mitigating negative effects on coastal ecosystems (*Burt et al., 2014*; *Canada British Columbia Marine Protected Area Network Strategy, 2017*). In nearshore habitats, methods that directly survey faunal communities have varying sampling efficacy across the complex seascape of structural habitats and environmental conditions. Monitoring fish biodiversity in the nearshore with visual or capture survey—such as underwater visual surveys or beach seining—is challenging when depths, substrates, and vegetation types differ and can lead to strong bias and very low detection probabilities for many species (*Liggins, Bradley & Kennelly, 1997*; *MacNeil et al., 2008*; *Emslie et al., 2018*). Traditional surveys also have logistical challenges of bringing trained personnel to sometimes difficult to access sites.

Environmental DNA (eDNA) metabarcoding surveys using water samples offer the potential to overcome many limitations of traditional survey methods and standardize biodiversity data collection across varying habitats (*Baillie et al., 2019*). However, because eDNA metabarcoding detects evidence of organisms' occurrences based on trace molecules, detection relies on processes of DNA release, decay, and advection from the source organism to the water sample that can vary across environments and organisms (*Kelly et al., 2016*; *Harrison, Sunday & Rogers, 2019*). For example, we expect detectability of a local (nearby) organism within an eDNA sample to increase with local abundance or biomass but decrease when environmental conditions confer faster decay or transport of particles from the sampled area (*Harrison, Sunday & Rogers, 2019*). By contrast, we might expect greater detectability of organisms from further distances from the sample when those organisms have high temporal variability in location (*e.g.*, shoaling behavior, in which distant species might have been present in high numbers just before sampling), or when environmental conditions confer higher transport, leading to greater arrival of particles from further afield. Although the influences of these factors have been considered to influence community assemblage patterns detected with eDNA metabarcoding, they have rarely been quantified (*Kelly et al., 2016*; *Harrison, Sunday & Rogers, 2019*; *Stewart, 2019*).

While the spatial scale of observation can vary due to unmeasured processes in eDNA surveys, it also has known and challenging effects on comparing estimates of biodiversity between methods (*Levin, 1992*; *Schneider, 2001*). Biodiversity patterns observed using a particular spatial extent of observation (*i.e.,* the sample grain) can be a property emerging from smaller-scale processes (*e.g.*, combined habitat availability), or larger scale processes (*e.g.*, climatic gradients) (*Levin, 1992*). Furthermore, dominant patterns observed at small scales might not persist when observed at larger extents and vice versa (*Wiens, 1989*; *Steele, 1991*; *Schneider, 2001*) (Fig. 1). Therefore, differences in sample grain specific to survey methods could influence which biodiversity patterns are observed and how we make ecological inferences to those patterns. Here, as in *Levin (1992)*, we use the term "scale" to refer to the spatial extent of an observation or process, such that larger scale refers to a larger spatial area. Although previous work comparing marine eDNA metabarcoding to traditional survey methods have shown general similarity among methods in community composition (*Aglieri et al., 2020*; *Polanco Fernández et al., 2021*; *He et al., 2022*), the spatial scale at which these similarities are maintained has not been fully explored (*Leduc et al., 2019*; *Robinson et al., 2023*).
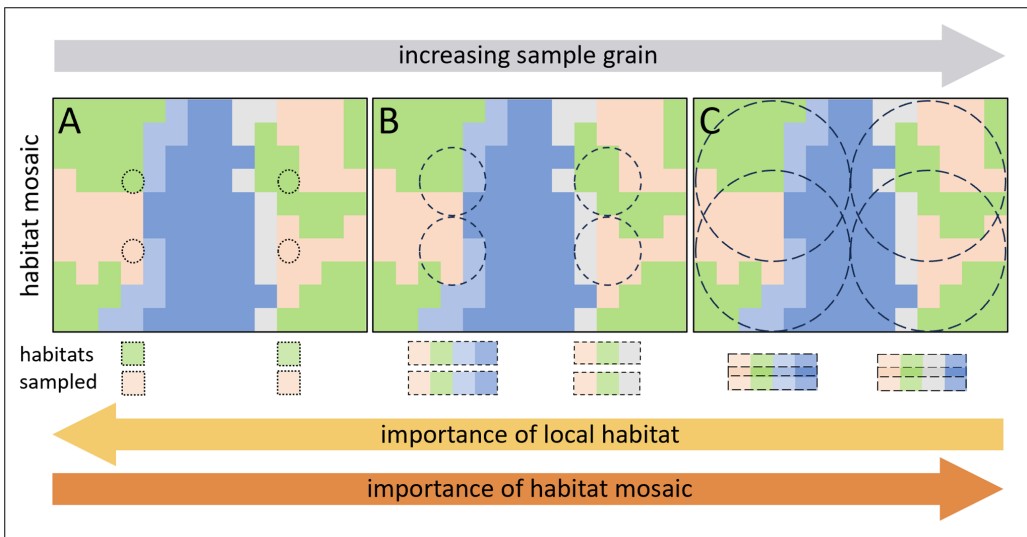

**Figure 1** **Hypothetical survey designs highlighting the interaction between scale of drivers and scale of observation.** Figure illustrates four sites sampled at different pairwise distances across which habitats and associated species change, with increasingly larger sample grains. (A) When sample grain spans only a single habitat, patterns in diversity associated with these habitats might be observed, even at short pairwise distances. Turnover of biodiversity between habitats sampled is observable (and expected) over both short and large distances. (B) When a sample grain spans multiple habitats, patterns in diversity associated with the centroid of the habitat may not be observed. Observed patterns (turnover) may reflect emergent effects of the habitat in the sampled area (sample grain extent), which might only occur when areas contain novel habitat features., *i.e.,* turnover is expected to be low. (C) When sample grains overlap, samples are homogenized and not independent. With increased sample grain size, the habitat mosaic within and unique to each sample also becomes increasingly more important in driving turnover of associated biodiversity.

A growing body of work has shown that patterns of fish community composition are detectable with eDNA metabarcoding in marine systems at "fine" or "small" spatial scales (*Port et al., 2016*; *O'Donnell et al., 2017*; *Jeunen et al., 2019*; *West et al., 2020*; *He et al., 2022*). In these studies, community compositional differences were typically assessed either across heterogeneous habitat types but within small spatial areas (100s to 1000s of meters; (*Port et al., 2016*; *O'Donnell et al., 2017*; *Jeunen et al., 2019*; *West et al., 2020*), or across larger spatial scales but considering a single habitat type (*e.g.,* seagrass beds; *He et al., 2022*). There has been a lack of studies that assess the effect of both habitat type and distance on compositional differences, which is needed to determine the scale at which eDNA can effectively detect compositional differences associated with habitat.

Relative to traditional survey methods, we expect eDNA to miss some low-biomass species detected within the fixed-area survey (*Port et al., 2016*; *He et al., 2022*), but to further gain some species missed by traditional methods or found outside the sample grain of the traditional survey, effectively sampling biodiversity at a greater but somewhat unknown spatial scale of inference.

Given the expected long duration of eDNA detectability in seawater (*Collins et al., 2018*) and empirical findings of high water mass fidelity (*Kelly, Gallego & Jacobs-Palmer, 2018*), it can be expected that the DNA present in a given water sample comes from
source organisms both in the immediate vicinity (*i.e.,* what might be defined as the "local biological community") as well as organisms present in more distant areas, especially when seawater movement is high. Thus, although the volume of water of an eDNA sample can be measured, the area across which source organisms of sampled DNA exist at the time of sampling cannot be directly measured and likely varies with water movement and other environmental and biotic conditions. Hence the eDNA sample grain (defined here as the area across which source organisms exist at the time of sampling) can be large compared to traditional sampling methods. This presents significant challenges to the interpretation of eDNA survey results, particularly when the goal is to understand patterns in species distributions, spatial turnover, and species richness (*Cristescu & Hebert, 2018*) (Fig. 1). By contrast, capture and visual survey methods tend to have fixed and known sample grains in both space and time but can miss the transient species that move in and out of the sampled space.

Despite reported similarities and differences between traditional survey methods and eDNA metabarcoding (*Port et al., 2016*; *Kelly et al., 2017*; *Jeunen et al., 2019*; *Aglieri et al., 2020*; *West et al., 2020*; *Polanco Fernández et al., 2021*; *He et al., 2022*; *Robinson et al., 2023*), little work has assessed the conditions and spatial scales that result in similarity between eDNA and traditional survey methods, nor how this affects biodiversity summary metrics such as richness and compositional differences. Assessing how the spatial grain of eDNA observations changes across spatially discrete habitats connected by water movement, and how the spatial grain of observations influences our ecological inference, can increase confidence in using eDNA metabarcoding surveys and provide a better understanding of the biodiversity patterns of fish communities that we recover from them.

Here we assess richness and compositional differences of species assemblages based on an eDNA metabarcoding survey using fish-specific primers and compare them to adjacent capture surveys. We do not compare absolute richness estimates as an evaluation of either method, because sampling effort could strongly affect the number of species captured, but instead ask how differences in richness vary across sites. For capture surveys, we use a common method for assessing nearshore fish diversity (*Magurran, Khachonpisitsak & Ahmad, 2011*; *Robinson, Yakimishyn & Dearden, 2011*; *Robinson & Yakimishyn, 2013*; *Iacarella et al., 2018*; *Proudfoot et al., 2023*) that uses a shallow net reaching from the surface to the seafloor, sampling fish within a fixed area with high detection probabilities given accurate species identification (*Parsley, Palmer & Burkhardt, 1989*). Like many other capture or visual survey techniques, this method has a small sample grain relative to many fishes' movements and local distributions (*Perry, Staveley & Gullström, 2018*; *Berkström et al., 2020*), is less effective on complex substrates and vegetation that foul the nets (*Baker et al., 2016*) and is logistically difficult to implement at large scales. By sampling fish assemblages using eDNA and beach seining across a nearshore marine seascape we assess the potential effects of both spatial distances and habitat differences between sites among methods. We also identify taxa, phenotypes, or environments that alter detectability in either method and identify the spatial scales at which eDNA can best identify patterns of changes in fish biodiversity in marine systems.

Using our paired sampling approaches, we test two hypotheses with specific expectations about how species characteristics, methods, and environments interact to influence detections. First, we test the hypothesis that species biomass influences eDNA detectability, predicting that among species detected in beach seines, those with lower biomass are more often missing in eDNA surveys. Second, we test the hypothesis that, due to seawater transport of eDNA, biodiversity detected in eDNA is influenced by factors occurring at a greater spatial scale than in beach seining. We predict that eDNA taxonomic richness is influenced by factors occurring at larger spatial scales than beach seining such that the number of taxa detected in eDNA relative to beach seining will increase with (i) proximity of samples to alternative habitats, (ii) surrounding habitat heterogeneity, and (iii) the degree of local seawater movement. We also predict that eDNA and beach seining detect different spatial patterns in biodiversity owing to different spatial scales of sampling and predict beach seining to yield patterns in fish biodiversity linked to habitat features that vary at smaller scales compared to patterns in biodiversity based on eDNA metabarcoding.

## MATERIALS & METHODS

### Study area

The Hakai Institute is an independent science organization funded by the Tula Foundation. Sampling by beach seining, and fish handling fall under the authority of Fisheries and Oceans Canada: Scientific Licenses XR2612014, XR512015, XR1262016, XR3172017, XR782018, XR1382019, and XMCFR192020 and BC Parks: Permits 107070 and 107090 for the Hakai-Luxvbalis Conservancy, collaboratively managed by the Province of BC, and the Haíɫzaqv and Wuikinuxv Nations. We followed Canadian Council on Animal Care guidelines on the care and use of fish in research, teaching, and testing (*Batt et al., 2005*) and were in compliance with sections 444 to 447 of Canada's Criminal Code. Captured fish were placed in aerated seawater inside rubber totes until both seine sets were complete. Seawater was exchanged regularly as needed. To reduce overcrowding and the chance of mortality, highly abundant, large, and easily identified species were counted and released immediately. After both seine sets were conducted, fish were taxonomically identified, measured for fork length, and released. As handling time was limited to <10 s per individual fish, no anesthetic was used.

We conducted paired eDNA and beach seine surveys at 18 sites over 3 years on the Central Coast of British Columbia (Fig. 2), territories of the Haíɫzaqv and Wuikinuxv First Nations resulting in 46 non-independent sampling events. The region is composed of small islands and archipelagos exposed to the open ocean (Queen Charlotte Sound) in the west and by fjords and channels of mainland British Columbia in the east. As part of Hakai Institute's nearshore ecological monitoring program, these sites were selected as representative of shallow-water soft-sediment habitats in the region. Sites ranged from small pocket beaches of soft sediment exposed only at low tide to large intertidal beaches and mud flats. Sites varied in their exposure to open ocean, vegetation type and density, sediment type, and seawater movement. Tides in this region are diurnal with typically morning low-low tides in the summer when research was conducted.

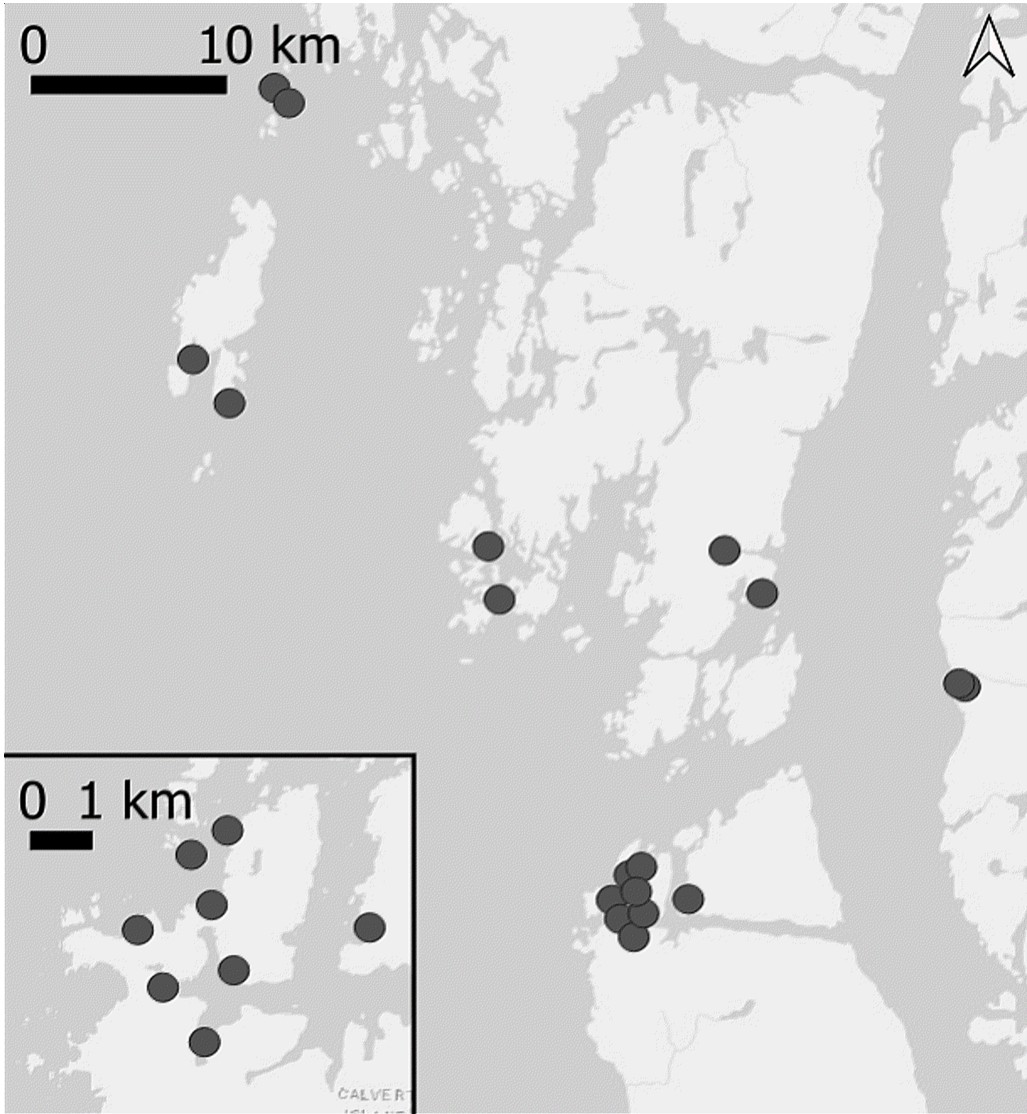

**Figure 2** Map of paired eDNA and beach seining survey sites on the Central Coast of British Columbia, 2018-2020.

## Beach seining surveys

Fish communities were surveyed at each site within 1 hr of morning low tide using beach seines set from a boat. All seine nets were 11 or 22 m long with four mm stretch mesh and a 3.1 m drop (height) in the center. During each survey, two seine sets were performed sequentially at opposite ends of each site to capture within-site variability—ranging from 10 to 100 m in along-shore distance, according to the size of the soft-bottom bed. Each set's survey area was approximated as a half circle (for the deepest extent of the seine net when the net was not yet extended) plus a rectangle (for the portion of the set where seine width was uniform). For abundant species a random sample of 20 individuals was retained for length measurements. After both seine sets were conducted, fish were taxonomically

identified, measured for fork length, and released. The abundance of each species at a site was estimated as the sum of the two seine sets. Biomass and biomass density estimates were made for each species at a site from the beach seine data (See Supplementary Text for more details). Biomass density for each species (grams per meter squared) was used to account for variability in the area covered by the seine sets which varied with beach morphology. Some of the species in the genera *Artedius, Citharichthys,* and *Sebastes* are difficult to identify to species level due to lack of differentiating morphological features, especially for juvenile stages within some genera. In these cases, we recorded the group of species within a genus to which we could be confident as "partial identifications". These partial identifications along with taxa identified to species are collectively the lowest identifiable taxa and are used throughout our analysis.

### eDNA surveys

Three replicate 1100-ml seawater samples were collected at each sampling site, located at the midpoint between each seine set, 30 cm below the surface. Over three years of sampling, we temporally coupled eDNA surveys to the beach seine surveys, most of which were sampled within 1 h to 2 days from the beach seine survey, spread out across the sampling season (spanning 14 days each year). However, in year 2, we conducted one single round of eDNA sampling at all sites within 2 days, these were 33 and 34 days after the beach seining samples were done in that year due to logistic constraints, and we therefore tested for effects of temporal offsets between surveys in our analyses. A total of 18 sites were visited across three years between 2018 and 2020 (Fig. S1 and Table S1). In 2018, 16 sites were sampled in July and six of these were resampled one month later to further assess the effects of temporal offsets in sampling. In 2019, 10 sites were sampled, and in 2020, 14 sites were sampled, and two of these sites were new to the dataset. All sites had matching beach seining data, resulting in 46 paired sampling events over three years. Each seawater sample was stored on ice in the dark for no more than 4 h until processing at the field station. In a sterile environment, we filtered 1000 ml of each replicate sample through a 0.22-um Sterivex filter unit™ using a peristaltic pump and sterilized tubing. We expelled sample water from filter housings, filled them with ∼2 ml Longmire's Buffer, and stored them in the dark at 4 °C. Sample processing and PCR amplification (See Supplementary Text for more details) was done at the Hakai Institute Genomics Laboratory. Bioinformatics and quality control are detailed in Supplementary Text. The taxonomic assignment was based on top blast hits of regional species (Alaska to southern California). A species annotation was retained when the best matches (based on % identity) were all to the same species. When multiple equal top hits (*e.g.*, 99.9% and 99.9%) from different species occurred we created a partial identification (Table S2, in a manner analogous to partial identifications in the beach seining surveys). Further details on taxonomic assignment can be found in Supplementary Text. Various approaches in cleaning metabarcoding data are used to address or remove potential detection errors, such as classifying singleton positive detections from a set of subsamples as false-positives (*Ficetola et al., 2015*) or occupancy modeling of sample replicates (*Lahoz-Monfort, Guillera-Arroita & Tingley, 2016*; *Dorazio & Royle, 2005*). We used occupancy models to remove possible false-positives by filtering out observations

of ASVs with <80% probability of occupancy based on the pattern of detections across sample replicates, that yields what can be considered a "high-confidence" dataset (*Kelly et al., 2017*) (see details of occupancy model fitting provided in Supplemental Information). However, because this can be considered a conservative approach, *i.e.,* retaining only ASVs with multiple occurrences at the possible cost of removing true positives, we present in the Supplemental Information select figures, summaries, and statistics, where the occupancy modeling step has been removed from the bioinformatics steps. This was done to weigh the effect that our data cleaning efforts had on our results. All other data processing and analyses remain the same as in the main text.

## Taxonomic resolution and habitat use

To aid comparability between beach seining and eDNA data, the partial and species identifications were made synonymous between eDNA and beach seining by grouping taxa based on the coarsest lowest identifiable taxa from either method, which we hereafter refer to simply as "taxa" (Table S2). This allowed direct comparison of methods even if species were not always fully resolved with either method. To assess if taxa detected by our methods were expected to be found in the habitats sampled, we categorized taxa based on their expected use (y/n) of marine shallow (<40 m) soft sediment habitat at any stage in their life-history according to published studies (*Hart, 1973*; *Allen & Smith, 1988*; *Lamb & Edgell, 2010*). For lowest identifiable taxa in which multiple taxa varied in expected habitats, we did not assign a category and omitted them from habitat-use analyses.

## Habitat variation and seawater movement

For each paired sample, we took temperature and salinity measurements at fixed depths along a transect perpendicular to shore. We recorded site characteristics: vegetation types and cover, slope, and sediment type (mud, sand, gravel, or cobble). To determine proximity to alternative habitats, we used GIS layers of habitat distributions from the British Columbia Marine Conservation Analysis (*The British Columbia Marine Conservation Analysis, (BCMCA), a Tides Canada Initiatives Project, 2008*; *The British Columbia Marine Conservation Analysis, (BCMCA), a Tides Canada Initiatives Project, 2009a*) and marine charts to derive distributions of discrete habitat types known to influence fish distribution (seagrass beds, kelp forests, rocky shore, deep water, and freshwater inputs) in the sampling region. We calculated the distance of each sample to each of these alternative habitats.

In lieu of direct measurements of seawater movement, we used the percent fine sediment (<64 µm) by weight as a predictor of seawater residence time (*Dean & Dalrymple, 2004*; *Davidson-Arnott & Greenwood, 2009*). Thus, a greater percentage of fine sediment by weight indicates less seawater movement, while a lower percentage of fine sediment indicates faster moving seawater with short residency. Sediment cores were collected at ~1.5 m (mean low tide), dried, and subject to grain size analyses to determine distribution of the sediment composition and calculate percent of sediment <64 µm at every site. We also used an exposure class GIS layer (*The British Columbia Marine Conservation Analysis, (BCMCA), a Tides Canada Initiatives Project, 2009b*) as a biologically-relevant indicator of wave exposure derived from fetch calculations (*Howes, Harper & Owens, 1994*).

## Analyses

We compared observations of taxa between methods at the regional scale (*i.e.,* all replicates, sites and sampling times combined, gamma-level) and at the local scales (*i.e.,* replicates combined within sites and years, alpha-level). To test the hypothesis that low-biomass species in the beach seines are more often missing from eDNA surveys (false-negatives) at the survey-wide scale, we fit a generalized linear model with a logit link function on survey-wide detections in eDNA (*i.e.,* presence *vs.* non-detection) as a function of maximum biomass density across the study. To assess the effect of local species biomass on eDNA detection at the site-level we visually compared biomass distributions of species detected *vs.* not detected at each site.

For species with non-zero detections in both methods in at least 33% of paired samples (15 out of 46), we tested for relationships between relative biomass in beach seines and read index in eDNA. Species with fewer detections across both methods were not analyzed to avoid analyses with inflated zeros. We fit linear regressions to relative biomass density as a function of read index and calculated R-square values. To test if taxa only detected in eDNA had different habitat preferences than taxa also detected with beach seining, we ran a binomial exact test on expected habitat use of species in each group, including all taxa for which we had a single expected habitat use.

To test the hypothesis that increased proximity to different habitats, habitat heterogeneity and seawater movement increases the number of taxa detected in eDNA relative to beach seining, we modeled taxonomic richness differences between methods as a function of these variables. We quantified differences in taxonomic richness for each paired survey event as the number of taxa detected in eDNA minus the number of taxa detected in beach seining. By modelling richness differences between methods as a function of these environmental variables rather than modelling the interaction between method and environment on absolute richness, we avoided singularities associated with complex and overfitted models. We used marginal models that allowed us to account for non-independence of sites that were repeatedly sampled 1-4 times over the three years of sampling. Marginal models allow the off-diagonal elements of the correlation matrix to be non-zero (*Liang & Zeger, 1986*; *Zuur et al., 2009*), so we can account for non-independence while avoiding the addition of random effects with low repetition within a group (*Pinheiro & Bates, 1996*; *Gelman & Hill, 2006*). We used a correlation structure that holds within-group correlations to be the same across time (*i.e.,* "exchangeable" structure) because differences in richness between methods are expected to be correlated at a site but not expected to change over time, given that "paired" sampling was always done within the same 2-month window.

To reduce the number of variables within a single model, we formulated hypotheses of the drivers of richness differences into seven different models that we could compare (Table S3). In the first three models we tested the effects of proximity to alternative habitats: seagrass beds (Zostera marina) and canopy forming kelp (model 1), rocky shoreline, deep water (>25 m deep), and subtidal slope (model 2) as well as freshwater (model 3; Table S4). We tested the effects of habitat richness (the number of different habitats) within a 100-m and 1000-m radius of the site (model 4), percent fine sediment (a proxy of seawater residence time; model 5) as well as the absolute number of days between seining and eDNA

sampling at a site (model 6). Lastly, habitat richness and fine sediment we tested in the same model (model 7). Because there were many variables and limited observations (46) we limited the total number of variables to three and used Akaike Information Criteria for small sample sizes (AICc) to identify the most parsimonious models. We considered models where ΔAICc <2 to be equivalent and report Akaike weights to represent the relative likelihood of each model (*Burnham & Anderson, 2001*). We used the variance inflation factor to test for multicollinearity among covariates within the same model. No variance inflation factors within the models exceeded two (*Billor, Chatterjee & Hadi, 2006*). We implemented this analysis in R v. 4.3.0 (*R Core Team, 2023*) using the package nlme v. 3.1 (*Pinheiro et al., 2017*).

To test the hypothesis that eDNA and beach seining detect different spatial patterns in nearshore fish biodiversity, we modeled compositional turnover from eDNA and beach seining surveys according to hypothesized predictors separately and compared the strength and importance of predictors. We used Generalized Dissimilarity Models (GDMs) to model the taxonomic turnover component of Jaccard's dissimilarity index (*Baselga, 2012*), as a function of over-water distance between samples, differences in collection date, and two environmental variables (vegetation cover and ocean exposure) that are known to affect nearshore fish distributions (*Robinson, Yakimishyn & Dearden, 2011*). We fit these models to beach seine and eDNA survey data separately. GDMs are an approach to modelling pairwise distances among sites against predictors that also take the form of site pairwise distances, that additively combines the effect of each predictor, constrains effects of each to increase monotonically, but allows non-linearities in their effects, and appropriately models error of values constrained between 0 and 1 (*Perry, Staveley & Gullström, 2018*; *Pinheiro & Bates, 1996*).

Both vegetation cover and site exposure are known to shape taxonomic turnover in nearshore fish communities (*Robinson, Yakimishyn & Dearden, 2011*), but they typically vary at different spatial scales. Vegetation cover typically varies in patchiness at scales of 10s to 100s of meters, whereas exposure varies at scales of 100s to 1000s of meters (Fig. S2). We calculated spatial distance between sites as over-water pairwise distance using QGIS software (*QGISDevelopment Team, 2023*). Within-site temporal replicate pairs were given an arbitrary pairwise distance of five meters (rather than 0) because each site visit was not sampled in precisely the same location due to GPS error and this vastly improved model fit. We used three (default) i-splines for each variable except pairwise distance, where we used four i-spines as it greatly improved the deviance explained by the model (*Ferrier et al., 2007*). We compared model fit, deviance explained, and significance of individual variables, and ran a model for each variable separately to estimate variance explained by that variable (*Ferrier et al., 2007*). To ensure that model differences between methods were not due to effects of taxa detected exclusively in either method, we also ran the models using only species detected with both eDNA and beach seining. We implemented this analysis in R using the package gdm v. 1.5.0 (*Fitzpatrick et al., 2020*).

## RESULTS

### Taxonomic summaries

Our beach seining surveys detected a total of 54 fish taxa identified to the species level, and three partial identifications—containing two, two, and three species each within three single genera, respectively—resulting in 57 unique taxa (Table S3 and Fig. 3). Our eDNA metabarcoding surveys detected a total of 59 species and 11 partial identifications—of which one partial identification contained seven, one contained three, and the remaining nine contained two species. This resulted in a total of 70 unique taxa. One partial identity from eDNA surveys contained multiple genera within the family Pleuronectidae, and the remaining were each within a single genus. Across both eDNA and beach seining surveys, we detected a total of 82 unique taxa, with 69 identified to the species level. eDNA and beach seining detected 44 common taxa, while 26 were exclusively detected with eDNA and 12 were exclusively detected with beach seining (Fig. 3).

### Species detection differences

Biomass densities of taxa detected in beach seines ranged across 12 orders of magnitude, from $1 \times 10^{-9}$ to 269 g/m$^2$ (see log scale in biomass density, Fig. 4). Taxa caught in beach seines but not detected with eDNA across the entire study made up 12 of the 23 lower biomass-density taxa by rank (Fig. 4, see also Fig. S3 for taxonomic assignments), their summed biomass densities represented less than 1% (0.14%) of total fish biomass, and the probability of detection by eDNA increased with maximum observed biomass density ($X_{21,N=56} = 21.36$, $p < 0.005$). This pattern was similar within sites, where fish taxa detected in the beach seines that were missed by the eDNA (eDNA false-negatives) were those with lower biomass density in the beach seines, both in terms of rank and absolute biomass density (Figs. 5A, 5B). Relationships between relative biomass density and read index in the eDNA at the site level was not strong (Fig. S4). Among the taxa for which we fit models the biomass-read index relationship (*i.e.,* those detected by both methods more than 33% of the time, $n = 29$), we found R2 values to be low (mean = 0.10, median = 0.04). Although it was not among our original hypotheses, we also found that the mean proportion of eDNA false-negatives at a site ($n = 18$) increased with mean number of taxa in the beach seine (Spearman's rank correlation, $r_{n=18} = 0.64$, $p = 0.004$, Fig. 5C).

Taxa detected in eDNA but not beach seine were often those with fidelity to hard-bottom substrates and deep water—habitats not directly sampled by beach seine surveys but were generally nearby (<2,000 m). We found that among all eDNA-only detected taxa, only 2 of 26 (7.5%, the arrow goby and spotted ratfish) have habitat preferences that include shallow soft sediments or seagrass, *i.e.,* the habitats sampled. Instead, taxa detected *only* in eDNA were more often classified as living in freshwater/estuaries, deeper, or hard-bottom habitats (24 of 26, or 92.5%). This differed from the 87% of taxa detected in both methods that were classified as preferring shallow and soft sediment habitats (binomial test, $p < 0.005$).

### Richness differences

Within each paired sampling event, there tended to be greater taxonomic richness detected using eDNA (mean number of taxa = 23.2, SD = 5.3) compared to beach seining (mean

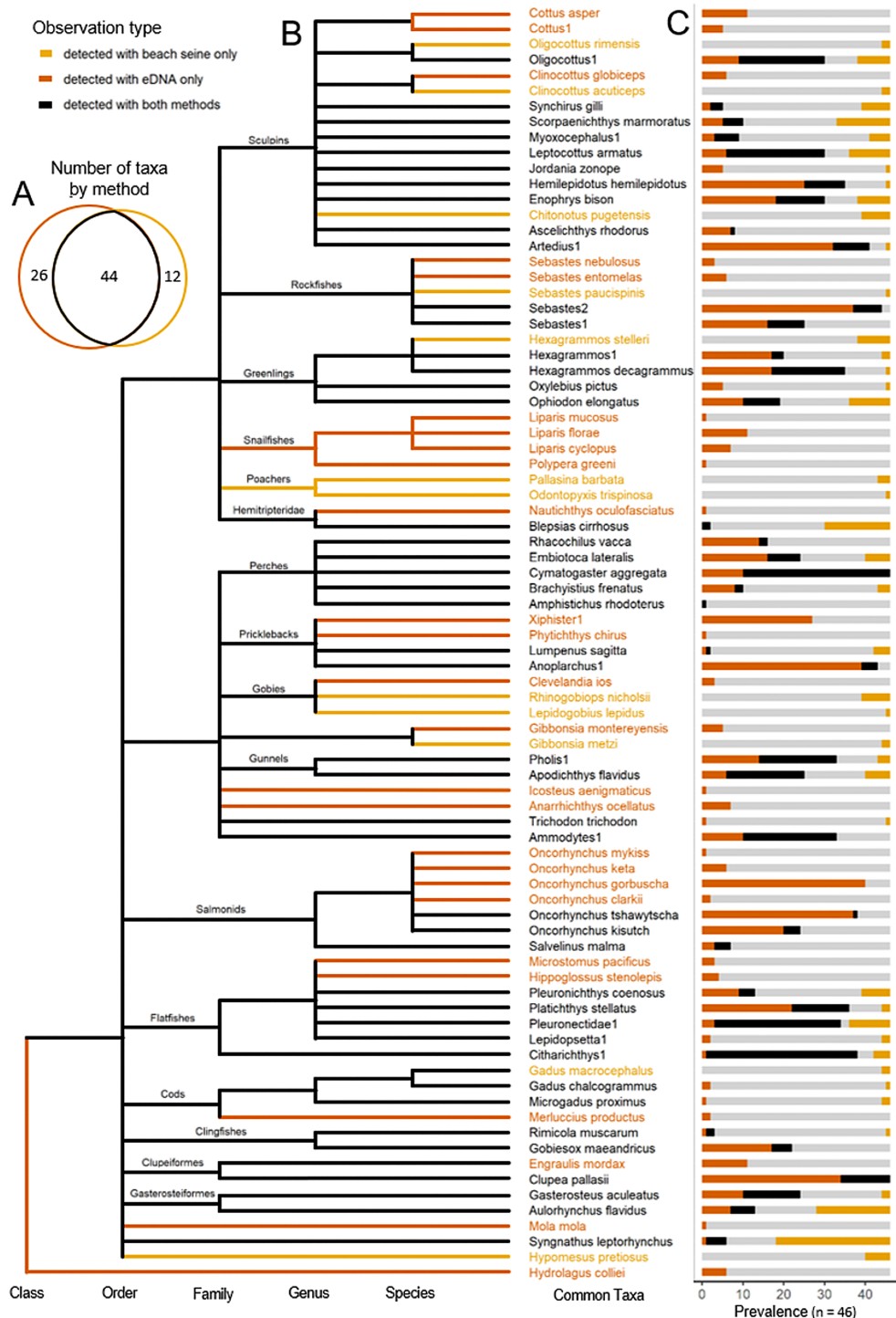

**Figure 3** Summaries of paired eDNA and beach seine surveys at 18 locations across 3 years on the Central Coast of British Columbia. (A) Euler plot of taxa detected (continued on next page...)

**Figure 3 (…continued)**
with eDNA only (orange), both methods (black), and beach seine only (yellow) for paired eDNA and
beach seine surveys at 18 locations across 3 years on the Central Coast of British Columbia. (B) Dendro-
gram of taxa by taxonomic ranking. Black lines indicate nodes and branches that were detected in both
methods. Orange and red lines indicate nodes and branches detected only in beach seine or eDNA surveys,
respectively. (C) Prevalence of taxa measured by eDNA only (orange), both methods (black), and beach
seine only (yellow). Grey indicates surveys where neither method detected a taxon.

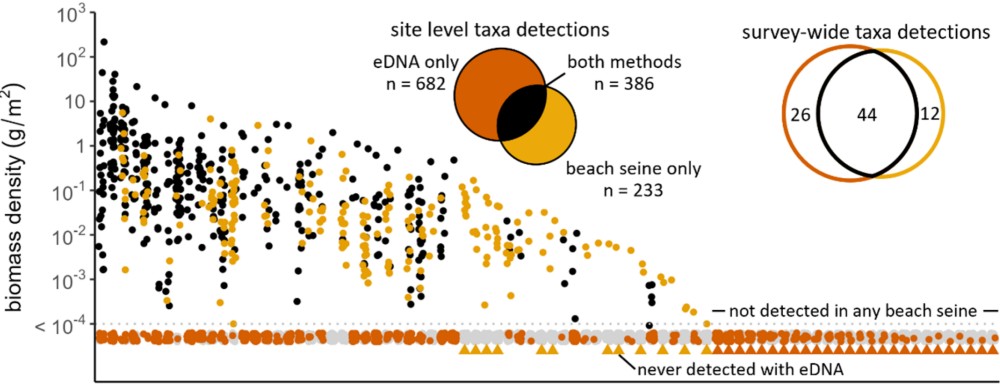

**Figure 4   Biomass density and detections from paired beach seine and eDNA surveys on the Central
Coast of British Columbia, 2018-2020.** Biomass density (grams/m$^2$) estimates of all taxa for each paired
survey. Biomass density is a summed across two beach seine set replicates. Black points indicate paired
surveys where the taxon was detected in both methods. Yellow and orange points indicate paired surveys
where a taxon was only detected in beach seine or eDNA surveys, respectively. Grey points indicate paired
surveys where the taxa was not detected with either beach seine or eDNA. Yellow and orange triangles in-
dicate taxa detected only in beach seine surveys and eDNA surveys, respectively.

number of taxa = 13.5, SD = 4.4) (Fig. 6). The top model that explained increased richness
in eDNA relative to beach seining contained variables describing seawater movement (%
fine sediment) and habitat richness within 100 m (Tables S3, S4, S5, and Fig. S5). The top
model accounted for 89.2% of the AICc weight (Table S5) and explained 27.8% of the
variance in richness differences. Percent fine sediments can be considered as an indicator
of seawater residence time, and greater percentages of fine sediment were associated with
smaller differences in eDNA richness relative to beach seining richness, in line with our
expectations (Fig. 6A). Further, eDNA richness was negatively correlated with percent
fine sediment (Spearman's rank correlation, $r_{n=18} = -0.58$, $p = 0.01$) but there was no
relationship with beach seine richness (Fig. 6A). Notably, temporal offset (absolute number
of days between paired eDNA and beach seine samples) accounted for less than 1% of the
AICc weight. Contrary to our expectations, habitat richness within 100m was negatively
associated with richness differences between the methods (Table S6). Increases in habitat
richness within 100 m was correlated with an increase in taxonomic richness in the beach
seine data (Spearman's rank correlation, $r_{n=18} = 0.50$, $p = 0.03$), but had no effect on
taxonomic richness in the eDNA (Fig. 6B).

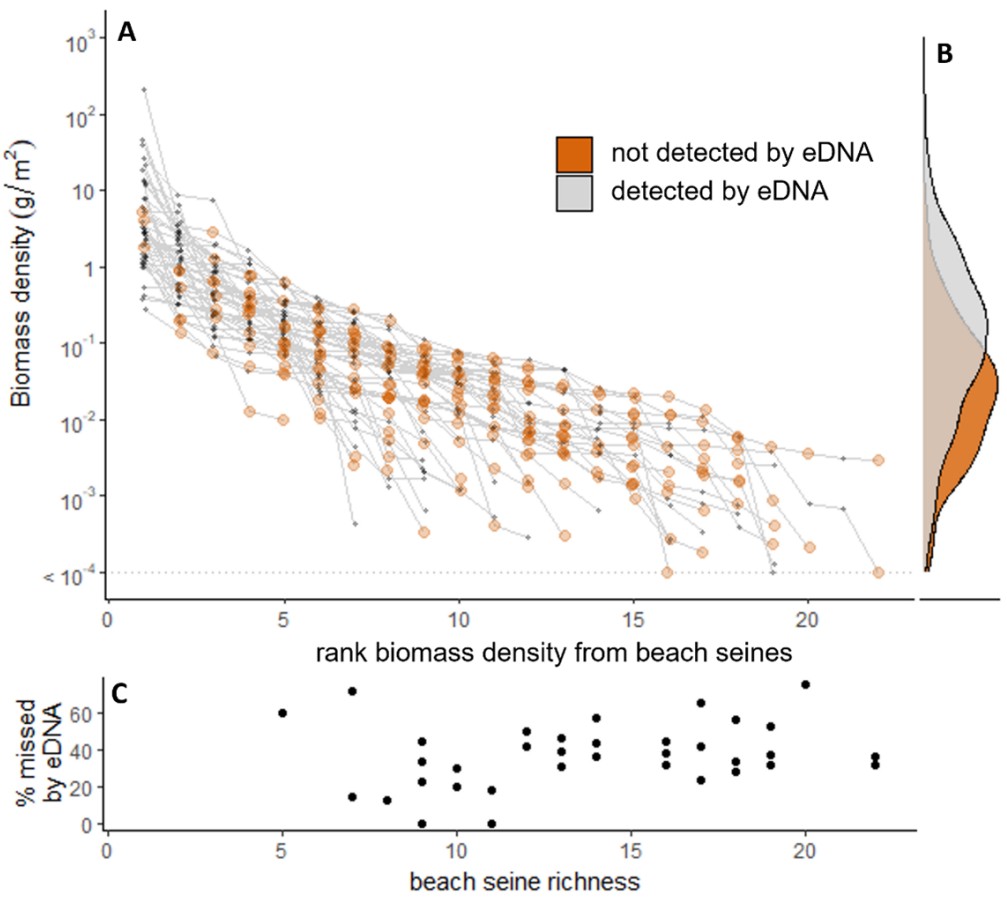

**Figure 5** **Relationship between rank biomass density of taxa within sites and positive detection in eDNA from paired beach seine and eDNA surveys of fish on the Central Coast of British Columbia, 2018-2020.** (A) Rank biomass densities of each beach seine survey. Lines connect individual surveys, point colour indicates taxa that were also observed with eDNA (eDNA true positives) *vs.* not observed with eDNA (eDNA false negative), right-most point indicates taxonomic richness of each seine along the *x*-axis. (B) Distribution of biomass density among eDNA true positives and eDNA false negatives. (C) Percent of taxa detected in beach seining but missed by eDNA (false negatives) as a function of beach seining richness for each paired survey. Each point corresponds to a terminal point (of a survey) in panel A.

## Spatial scales of turnover

Community turnover among pairs of samples was lower on average in the eDNA (mean = 0.43, SD = 0.14) than in beach seining (mean = 0.58, SD = 0.17; paired *t*-test, t(1034) = −24.90, $p < 0.001$), and the pattern of variation in pairwise turnover in eDNA and beach seining was weakly but significantly similar (Mantel Statistic $R^2 = 0.23$, $p < 0.001$). Principal Coordinate Analysis (PCoA) of turnover showed that eDNA had stronger spatial clustering compared to beach seining (Fig. 7). By contrast, the variation among sites in beach seine assemblages were not resolved at the subregion scale but appeared to be explained better by local habitat features, such as vegetation cover, that varied within subregions (Fig. 7B).
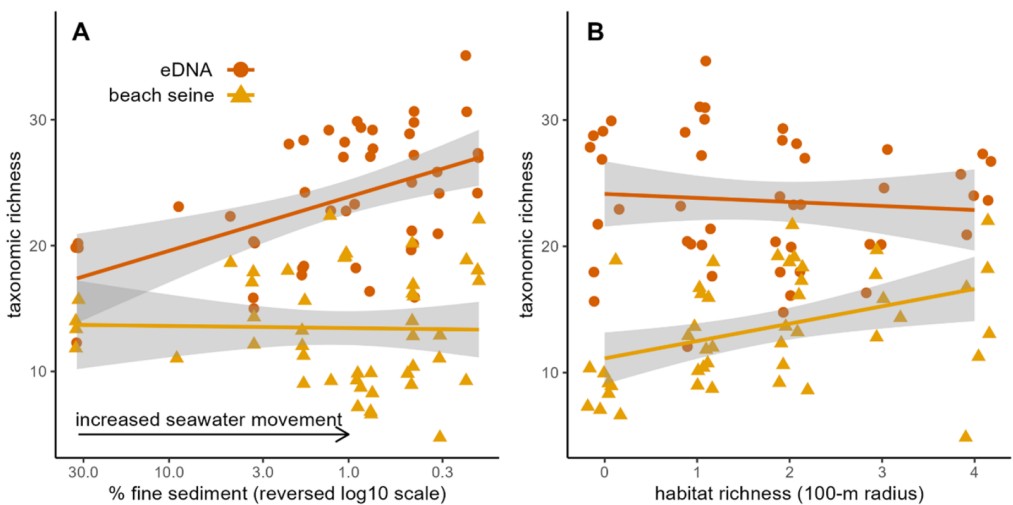

**Figure 6** **Taxonomic richness of paired beach seine and eDNA surveys over gradients of silt percentage and habitat richness within 100-m radius on the Central Coast of British Columbia, 2018-2020.** Circles indicate eDNA richness, triangles indicate beach seining richness. Lines represent post-hoc linear regression models of eDNA and beach seine richness and shaded areas show standard error. Silt percentage is used here as an inverse indicator of water movement.

GDMs of taxonomic turnover fit to the data detected by each method revealed key differences in how space and habitat features explain community turnover among methods (Table S7). In the model of eDNA-based community turnover, distance between sites was the strongest predictor of turnover, explaining 20% of the deviance in turnover, and predicting an increase in turnover with distance up to ~10 km with greatest turnover of ~25%. Beyond 10 km of pairwise distance, no additional turnover was associated with distance between sites (Fig. 8). Second was ocean exposure-level, in which exposure differences between sites explained 14% of the deviance in turnover. Vegetation cover difference was the weakest of all predictors of turnover in the eDNA-based model and was not found to be significant. In the model of beach-seining based community turnover, vegetation cover difference significantly explained most (13%) of deviance in turnover and predicted a total of ~37% turnover across the entire gradient. Exposure difference was also a significant predictor of turnover in beach seining models, with similar summed i-spline shapes and total turnover as in the eDNA model. Unlike the eDNA model, distance between sites was not a significant predictor of turnover, and only a small and uncertain percent of turnover was associated with very large distances (60 km) between sites. The number of days between surveys was not a significant predictor of turnover in either eDNA or beach seining models. The model results were similar when we ran the models with only taxa that were regionally detected with both methods ($n = 44$), indicating that findings were not driven by method-specific differences in detections (Table S8).
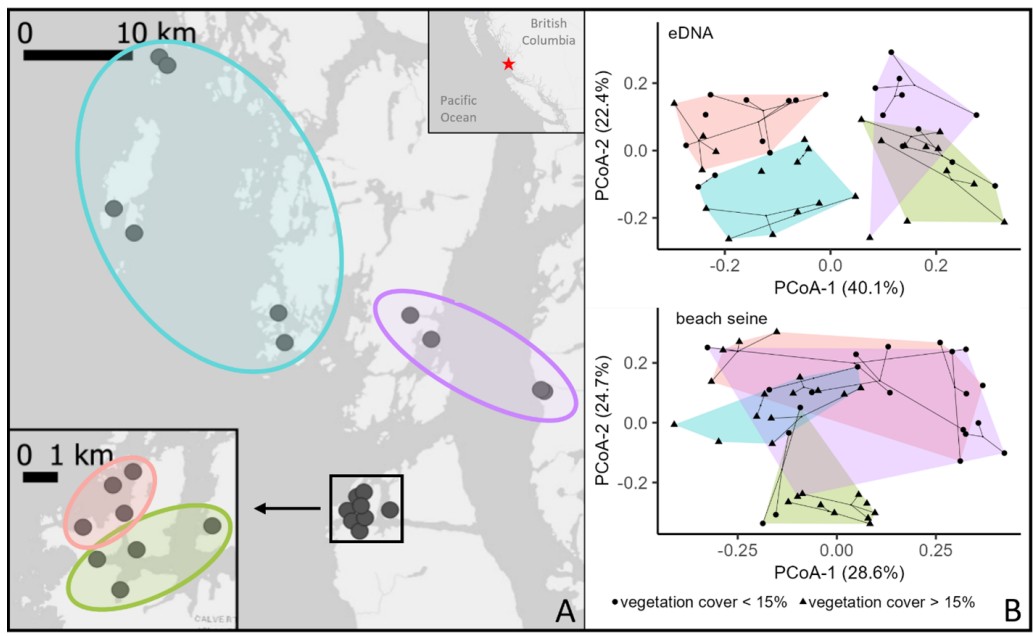

**Figure 7 Map and PCoA plot of paired eDNA samples and beach seining samples on the Central Coast of British Columbia, 2018-2020.** (A) Map of locations. (B) PCoA plot of taxonomic turnover component of Jaccard's dissimilarity index from beach seining and eDNA surveys on the Central Coast of British Columbia. Ellipse colour in (A) and polygons in (B) refer to sub-regions within the study area: Northern-exposed (blue), Northern-protected (purple), Southern-exposed (orange), and Southern-protected (green).

## DISCUSSION

Although the sampling grain of eDNA was variable, we found substantial evidence that it was larger than that of beach seining. Our findings show that fish biodiversity from eDNA metabarcoding samples can resolve taxonomic turnover at large spatial scales, but not vegetation-related variation at smaller scales as could be found with beach seining. We also found evidence that the sample grain of eDNA varies with seawater movement, as taxonomic richness in eDNA samples was greater relative to paired beach seining surveys in sites where sediments indicate higher seawater movement. Additionally, we found eDNA detectability to be sensitive to the relative local biomass density of taxa (*Port et al., 2016*; *Yates et al., 2021*; *He et al., 2022*); the species eDNA missed were on average those with lower biomass, which were more prevalent when local richness was higher. These findings suggest that eDNA surveys detect spatial turnover patterns across environmental gradients known to influence fish distributions at mid- to large scales but not at smaller scales, suggesting eDNA has a larger sample grain than beach-seining surveys. Our results build on previous efforts to define the spatial resolution of marine eDNA samples (*Port et al., 2016*; *O'Donnell et al., 2017*; *Kelly, Gallego & Jacobs-Palmer, 2018*; *Jeunen et al., 2019*; *West et al., 2020*; *He et al., 2022*) by highlighting the spatial scales at which we can make ecological inference, and identifying the conditions that influence it. Here we discuss our findings on missing low biomass species, the evidence for larger and varying sample
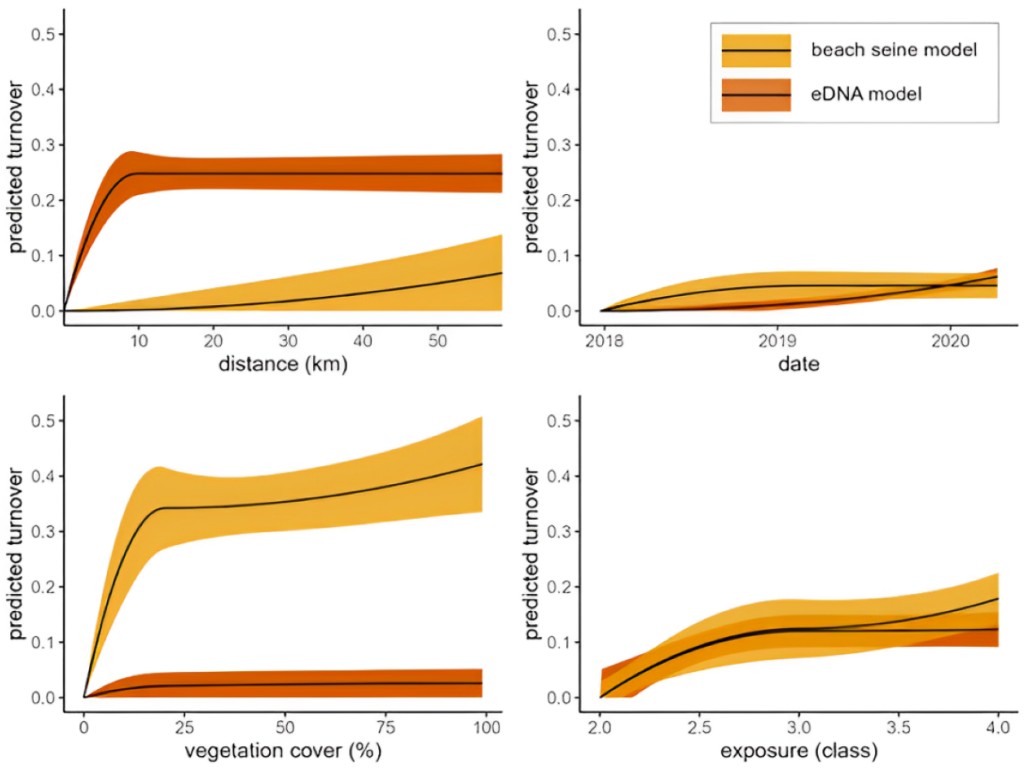

**Figure 8 Taxonomic turnover associated with distance, days between sampling, and difference in vegetation cover and exposure class for eDNA and beach seining surveys, on the Central Coast, BC, 2018-2020.** Lines with error bands (+/- 1 standard deviation) represent i-splines of partial effects of factors across their gradients. Variables were (pairwise distance-over-water), physical ocean exposure class, vegetation percent cover, and Julian data beginning on January 1, 2018. Except for spatial distance, the slope between any two locations along the line indicates the estimated rate of turnover across their range, *e.g.*, beach seining turnover in response to vegetation cover is estimated at around 0.35 (or 35%) between sites with 0% and 25% cover, and near 0 between sites with 25% and 50%. With spatial distance however, rates of turnover are always assessed starting at the origin –one site at the origin and another at a certain distance along the *x*-axis.

grains in eDNA, and the effects of different sampling methods on community turnover. We further discuss how eDNA metabarcoding could be used either on its own or to complement other methods of surveying fish biodiversity.

Our analysis reaffirms that species with lower local biomass density are more likely to be missed by eDNA (*Port et al., 2016*; *Yates et al., 2021*; *He et al., 2022*). Our results further highlight the nuanced issue that arises due to typical abundance distributions across natural communities—that the proportion of low-biomass species increases when local richness is higher, and therefore more are missed by eDNA (*i.e.,* increased false-negatives). A more taxonomically-rich community is predicted to have a higher proportion of rare (or low biomass) species as resources are partitioned amongst a greater number of species (*MacArthur, 1970*) and also may be found in more habitat-rich areas (*Dornelas et al., 2009*) where more but smaller habitats are packed, leading to lower biomass of supported species. As such, the proportion of species richness missed by eDNA likely increases with richness
itself, a phenomenon also found in other survey methods (*Palmer, 1990*; *Boulinier et al., 1998*; *Brose, Martinez & Williams, 2003*). The consistently lower biomass of species missed by eDNA is reminiscent of Preston's Veil—defined as a critical abundance level in surveys below which species go undetected (*Preston, 1948*). Increased sampling effort can shift the veil line to expose more rare species (*Preston, 1948*). Perhaps by analogy, the false-negatives in eDNA sampling associated with low biomass density of fish can be reduced with greater seawater sampling and/or deeper sequencing.

Although most of the study-wide eDNA false-negatives (never detected with eDNA) had among the lowest biomass densities, some high biomass taxa were occasionally missed with eDNA at some sites but not at others (Figs. 3, 5 and Fig. S7). These high-biomass false-negative detections suggest low or variable detection probabilities, possibly connected to poor PCR affinity (*Lamb et al., 2019*; *Van der Loos & Nijland, 2020*), environmental influences at the sampling site (*Harrison, Sunday & Rogers, 2019*), or chance events during the sampling and PCR process (*Kelly, Shelton & Gallego, 2019*). The solution to reduce missed high biomass taxa may simply be to increase sampling effort through increased field sample replications, primer sets, PCR replications, or sequencing depth.

We found evidence that seawater movement was associated with increases in eDNA richness relative to local richness (Fig. 6A), suggesting that influx of seawater from elsewhere can inflate local species richness as detected using eDNA. This finding supports previous theoretical work in which the role eDNA transportation was hypothesized (*Barnes & Turner, 2016*; *Cristescu & Hebert, 2018*; *Harrison, Sunday & Rogers, 2019*), but represents the first test of this hypothesis in marine systems to our knowledge. This finding suggests that the grain of an eDNA sample can vary across the seascape–increasing in areas of higher seawater movement.

Although the sampling grain of eDNA was variable, we found substantial evidence that it was larger than that of beach seining. There were generally more taxa detected with eDNA than with beach seining, and the taxa unique to eDNA appeared to be from adjacent habitats. The taxa that were only detected with eDNA were mostly classified as preferring habitats other than shallow soft sediments, supporting the hypothesis that the eDNA sample represented a broader range of habitats. Some of the species detected only with eDNA (*e.g.*, salmons and anchovy, Fig. 3) were possibly occasional visitors, as they would be expected to move in and out of the target habitat but easily missed with beach seining if the narrow temporal sampling event did not overlap with those species' occupation of the site. If so, this would imply that eDNA also has a broader temporal sampling grain (*Nielsen et al., 2007*; *Barnes et al., 2014*; *Collins et al., 2018*), which could be further investigated. We note that increased sampling with either beach seining or eDNA could influence species richness and could increase the number of extra-habitat species detected by both methods, so an important caveat is that our findings are relative to the sampling intensity as set in our study.

Our finding that taxonomic richness in beach seines, but not in eDNA, increased with habitat diversity within 100 m was contrary to our hypothesis. However, a heightened local taxonomic richness in areas of high habitat diversity in beach seining was previously found by Proudfoot (*Proudfoot et al., 2023*), and the effect was only detected at 100 m, but

not 1000 m, suggesting fish in beach seines either 'spill' out of their preferred habitat or there is some effect of habitat on local productivity at small spatial scales. The lack of an effect of habitat richness on species richness in eDNA could be due to its larger sample grain, that does not detect changes over small-scale environmental gradients within the extent of eDNA homogenization. Our analyses of community turnover across space and habitats indicate that scales of ecological inference are larger with eDNA than with beach seines. The importance of small-scale habitat variability (10s–100s of meters) on taxonomic turnover observed in the beach seining results is similar to findings in other capture-based studies of nearshore fish distributions in the Northeast Pacific, where vegetation cover or vegetation density were found to be the main drivers of species turnover (*Robinson, Yakimishyn & Dearden, 2011*; *Iacarella et al., 2018*; *Proudfoot et al., 2023*). The inability of eDNA sampling to resolve these small-scale differences could be explained by eDNA being physically mixed or homogenized through water and species movement at a scale larger than variation in vegetation (Fig. 1). The larger-scale environmental variable of exposure, which varies at scales of 100s to 1000s of meters, was an important predictor of fish species turnover in both eDNA and beach seining in both strength and form (Fig. 8), and eDNA detected distinct community differences at large spatial scales (Fig. 7B). Therefore, while eDNA may not detect the same small-scale differences in turnover as can be observed with beach seining, it does capture turnover at larger scales.

Distance between sites had a greater effect on community turnover in eDNA compared to beach seining and may be better understood by considering small and large distances separately. At small distances, eDNA samples were more similar, possibly owing to nearby eDNA samples having sample grains that span a similar set of habitats, or to eDNA being physically mixed or homogenized across sample locations (similar but slightly different processes; Fig. 1). For example, the sub-region 'southern-exposed' contains four sites within 4 km of one another of similar exposure but that differ greatly in their vegetation cover (0–70%). With beach seining, these sites were found to be among the most different across the entire region, but they were relatively similar with eDNA (Fig. 7B). This does not mean that differences cannot be found at small scales using eDNA (*Port et al., 2016*; *O'Donnell et al., 2017*; *Jeunen et al., 2019*; *He et al., 2022*), but that they may not be independent due to seawater homogenization—making interpretation of local-scale variation difficult (*Leduc et al., 2019*; *Robinson et al., 2023*). At large distances, greater community dissimilarity in eDNA compared to beach seining was possibly due to unmeasured environmental or biogeographical factors that affect the larger set of species detected at a site with eDNA, such as temperature, salinity, or historically different regional species pools. Hence, some factor other than exposure likely contributes to the distinct community differences we found with eDNA across major sub-regions, which were not reflected in the beach seining samples (Fig. 1B).

We found that the temporal offset of paired sampling events did not strongly influence richness differences between eDNA and beach seining—reinforcing the validity of our findings. Most paired sampling occurred within 2 days of each other, however, due to logistic restraints in 2019, beach seining and eDNA sampling occurred 33–34 days apart.

Although seasonal differences in local richness are expected, our analysis suggests that local richness remains somewhat constant during summer months.

Based on prevalence differences of taxa between methods, detection biases likely differ between eDNA and beach seining (Fig. 4 and Fig. S7). Beach seining, because of its small spatial and temporal sampling likely tends to miss species that form dense schools and species that regularly move in and out of the sampled habitat. For example, sandlance, herring, juvenile salmons, or northern anchovies (examples of dense schoolers, (*Hart, 1973*; *Lamb & Edgell, 2010*), and salmonids, greenlings, and surfperches (examples of movers, (*Hart, 1973*; *Lamb & Edgell, 2010*) were detected more frequently (or exclusively) with eDNA (Fig. 3 and Fig. S7). Additionally, when targeting seagrass beds and soft sediment habitat, beach seines also miss fish species from immediately adjacent shallow water habitats. Most notably would be habitats containing boulder or bedrock substrates that host taxa rarely detected over soft sediments (Fig. 3 and Fig. S7), such as some of the sculpins (Cottoidea), snailfishes (Liparidae), Pricklebacks (Stichaeidae), and Gobies (Gobiidae), which often inhabit void spaces in the substrate (*Hart, 1973*; *Lamb & Edgell, 2010*). Our results suggest that species with such distributions and behaviors may be better detected using methods such as eDNA being able to detect residual eDNA from species that passed through a targeted habitat, or species inhabiting substrates that are not so easily sampled with a beach seine (Fig. 3). With eDNA however, in addition to field sample replications, primer sets, PCR replications, and sequencing depth, size, behaviour, shedding rates, reproductive stage, and primer binding efficiency likely play a role in where and how much eDNA is sampled and how well it is amplified, *i.e.,* whether it is detected or not (*Harrison, Sunday & Rogers, 2019*). These differing methodological biases of either method, although making direct comparisons of survey results difficult, provide complementary information on species distribution where the alternate method may come short, and give a more complete assessment of gamma, alpha, and possibly beta-diversity.

Beach seining and eDNA provide two complementary views of the mechanisms shaping biodiversity in coastal marine systems due to the difference in their sample grains. Although beach seining is less effective (subject to increased escapement) outside shallow, soft sediment systems due to logistical and physical constraints (*Liggins, Bradley & Kennelly, 1997*; *MacNeil et al., 2008*; *Emslie et al., 2018*), it provides insights into biodiversity changes over small-scale changes in the environment—one of the scales important to both protection planning and monitoring (*Pante et al., 2006*; *Foley et al., 2010*; *Burt et al., 2014*; *Canada British Columbia Marine Protected Area Network Strategy, 2017*), as well as our understanding of the small-scale mechanisms shaping biodiversity in this region (*Robinson, Yakimishyn & Dearden, 2011*; *Iacarella et al., 2018*; *Olson et al., 2019*; *Proudfoot et al., 2023*). Sampling and sequencing eDNA, not limited to specific habitats, may achieve similar objectives, but with inferences made at larger spatial scales and across a greater diversity of habitats, though may lack information about responses to smaller subunits of environmental heterogeneity across which eDNA homogenizes.

Our study shows that eDNA metabarcoding differs from traditional methods for sampling nearshore fish communities by expanding the spatial scales at which we observe biodiversity. Because our work indicates that the sample grain of eDNA can vary with water

movement, future work will benefit from better understanding of how this effect could be corrected or accounted for in sample designs, analyses, and choice of beta-diversity metrics. Our work shows that eDNA metabarcoding generally misses species of low local biomass, leading to an often undetected 'veil' over low-abundance taxa. Pairing eDNA surveys with other survey methods with known detection probabilities for at least a subset of sample locations can help estimate the rate of eDNA false-negatives in a study, and increasing eDNA sampling effort can likely help remove them. Although the large sample grain of eDNA imposes a lower limit on ecological inference, the ease with which sampling may be expanded across a much broader range of habitats makes eDNA a potentially powerful tool in monitoring marine fish biodiversity at large spatial and temporal scales.

## ACKNOWLEDGEMENTS

We would like to thank the Haíɫzaqv and Wuikinuxv Nations on whose land this work was conducted. We are grateful for the technical support from the Hakai Institute and Tula Foundation. We are also grateful for the support of field and lab technicians at the Hakai institute. Notably, Midoli Brescht, Zach Monteith, and Angeleen Olsen for their help conducting beach seining and water sample collection and Rose-Lynn Savage, Carolyn Prentice, and Kristin M. Robinson for their help in the genomic lab.

### Funding

This work was supported by Natural Sciences and Engineering Research Council of Canada (NSERC). The funders had no role in study design, data collection and analysis, decision to publish, or preparation of the manuscript.

### Grant Disclosures

The following grant information was disclosed by the authors:
Natural Sciences and Engineering Research Council of Canada (NSERC).

### Competing Interests

The authors declare there are no competing interests.

### Author Contributions

- Ben Millard-Martin conceived and designed the experiments, performed the experiments, analyzed the data, prepared figures and/or tables, authored or reviewed drafts of the article, and approved the final draft.
- Kate Sheridan analyzed the data, authored or reviewed drafts of the article, and approved the final draft.
- Evan Morien analyzed the data, authored or reviewed drafts of the article, and approved the final draft.
- Matthew A. Lemay conceived and designed the experiments, authored or reviewed drafts of the article, and approved the final draft.

- Margot Hessing-Lewis conceived and designed the experiments, authored or reviewed drafts of the article, and approved the final draft.
- Rute B.G. Clemente-Carvalho performed the experiments, authored or reviewed drafts of the article, and approved the final draft.
- Jennifer M. Sunday conceived and designed the experiments, performed the experiments, prepared figures and/or tables, authored or reviewed drafts of the article, and approved the final draft.

## Animal Ethics

The following information was supplied relating to ethical approvals (i.e., approving body and any reference numbers):

Fisheries and Oceans Canada.

## Field Study Permissions

The following information was supplied relating to field study approvals (i.e., approving body and any reference numbers):

Haíɫzaqv and Wuikinuxv Nations and Fisheries and Oceans Canada approved field work.

## DNA Deposition

The following information was supplied regarding the deposition of DNA sequences:

The sequences amplified from water samples are available at NCBI under BioProject ID PRJNA1051585: BioSamples: 39222712–39222995.

## Data Availability

The bioinformatics scripts are available at Zenodo: morien. (2023). hakaigenomics/calvert_12S_methods_2018-2020: Methods for Calvert 12S eDNA, 2018-2020 (bioinformatics). Zenodo. https://doi.org/10.5281/zenodo.10183046.

The beach seining data, post-bioinformatics eDNA data including ASV table and annotation table, and analysis scripts are available at Zenodo: Benmm. (2024). Bennymm/-marine_fish_eDNA_beach_seine_central_coast: Data and code for eDNA/beach seine analysis (v1.0.0). Zenodo. https://doi.org/10.5281/zenodo.12688621.

## Supplemental Information

Supplemental information for this article can be found online at http://dx.doi.org/10.7717/peerj.17967#supplemental-information.

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
