# Peer review of "Effect of environmental DNA sampling resolution in detecting nearshore fish biodiversity compared to capture surveys"

_PeerJ, doi:10.7717/peerj.17967_

## Round 0.1 · original submission · Major Revisions

Three colleagues have reviewed your manuscript and recognized the research work you have carried out. On the other hand, they pointed out problems that make the manuscript unacceptable in its current form (see reviewers' comments for details).

I hope that the reviewers' constructive comments will enable you to revise the manuscript substantially, which is a prerequisite for its acceptance.

·

Basic reporting

no comment

Experimental design

no comment

Validity of the findings

it is not clear where the eDNA fastq files, ASV table, scripts for modeling and diversity analyses are deposited.

Additional comments

A few minor weaknesses that need attention before publication:
• Some sentences are overly long and could be rephrased for better clarity, for instance, lines 161-164 and several others, especially in the introduction.
• Regarding line 119-121 in the introduction, while it's stated that the grain size cannot be directly measured, it may be possible to measure the amount of water filtered for eDNA using a flowmeter within a given time period.
• Genera names should be italicized in line 203-204.
• Clarification is needed on whether 1000ml of each sample refers to the total from the three replicates or the entire sample in line 223.
• In line 349, specify whether the "54 fish" detected by beach seining refers to 54 species or 54 individual specimens
• Define how species are determined in the eDNA approach and the criteria for identifying unique taxa in eDNA when using the term "species" in line 352.
• Clarify the reduction from over 900 ASVs initially detected to 107 ASVs mentioned in the supplementary text.

Reviewer 2 ·

Basic reporting

see general comments

Experimental design

see general comments

Validity of the findings

see general comments

Additional comments

General comments

1. Overall, the scientific merit of this study is the assessment of different spatial scales in detecting fish diversity among intertidal habitats and differences between methods. However, it was not clear how hypotheses 2 and 3 differed sufficiently and needs to be better scoped. Also, I have outlined several key limitations of the existing study below that make testing of the hypotheses problematic.
2. Most importantly, the direct comparison of the eDNA results with the beach seine results (e.g., detections) is not valid. Although the authors claim to have used a common method of other studies (Line 140; which should be in the methods not the introduction section) they did not follow the standard approach of 3 beach seine sets. Using only two sets in an expected low diversity habitat (soft sediment beaches are well-known to harbour less fish diversity compared to kelp, eelgrass or algal dominated areas) and would not generate sufficient/adequate information on local fish diversity; based on my 25 years of experience, even using 3 seines in highly structured intertidal habitats would still only capture 80-90% of local coastal fish diversity.
3. Related to the inadequate seine sampling issue is assessing eDNA from triplicate water samples and assuming that only one DNA hit of three samples confirms presence of a fish species. The authors should refer to Cole et al. 2021 (Estuarine fishes associated with intertidal oyster reefs characterized using environmental DNA and baited remote underwater video) for a more tractable approach to using eDNA from water samples in detecting local fish diversity. Also, in this study and according to the authors, 24 of 26 species detected by eDNA were “not expected” in these nearshore habitats (authors table s1), and hence the data over-represent fish diversity detected at local or regionally by eDNA. It is also key that a table of the fish species caught and their detections by each method be included within the MS and not as a supplementary table; this is important information for the reader to have at hand in interpreting the results. Overall, the point is that any direct comparison of fish diversity between seining and eDNA is not valid and not adding any new information. Note however, that this doesn't preclude a comparison of model results from hypothesis 2/3.
4. Throughout the MS there needs to be some clarification of terminology used. For instance, the use of spatial scale terms is well founded in geography and is applicable to this study. However, throughout the MS the authors have flipped the definitions of large and small scale (which can be confusing to those who understand the correct usage). Large scale refers to local areas with higher detail, while small scale refers to regional areas with less detail. Hence, information associated with a specific beach is by definition large scale, while distribution of species along the coast is small scale. The authors should stick to convention, otherwise there will be unnecessary confusion.
5. The authors use “biomass density” which is unusual term in the marine literature. Is this being calculated for the eDNA results? And/or the seine results? If the latter, the two sets cover the same surface area of a beach and why would a density measure be needed. The authors also use biomass of fish which is sufficient in this study; why is the lack of detection by eDNA only considered relative to low fish biomass…what about species shedding rates, and other factors? Other factors should be discussed.
6. The term “seawater turnover” (line 251) is not applicable to nearshore coastal (intertidal areas) and is rarely used in marine studies except perhaps in studies of basins or inlets. The authors also use “seawater movement” which is more appropriate and 'measured' using their shoreline exposure index (Not sure how useful the sediment information is, if an exposure index is used. For instance, there would presumably be an established relationship between high exposure and coarser intertidal sediments and low exposure and finer sediments.)
7. Lines 177-190. Why is an ethics statement in the middle of the methods section? Should be placed at the end of the MS.
8. “we therefore tested for effects of temporal offsets between surveys in our analyses.” Where are the results of this analysis and why isn’t it discussed?
9. Discussion comments: L438/39. What were the low biomass fish species not detected by eDNA? Not all low biomass fish species are necessarily rare. Some fish species like kelp clingfish are small-bodied and numerically abundant and likely have very low shedding rates; other reasons why low biomass species were not detected by eDNA should be discussed.
10. The result that some high biomass species were not detected by eDNA at some sites but not others is also key (L471/72). What is curious is that the discussion revolves around issues like PCR affinity but doesn’t consider the ecology of the fish species; does the life history of the fish, behaviour, shedding rate, type of scales, etc have no influence on eDNA detections? This is a major weakness of the study as sampling fish biodiversity in the real world it is not simply a lab problem!
11. L481. “...inflate local species richness as detected using eDNA”. This is an important result and perhaps would also benefit from using Cole et al. (2021) approach as to how many water samples was a species DNA detected. If there is only one detect of three water samples of a species from offshore (eg halibut) then it is not a local species vs a 3 for 3 detects of species x.
12. L492. “…did not overlap with the species’ occupation of the site”. But this again points to a need to understand and report that species like anchovy and salmon are highly mobile, schooling species that may only occur along a beach for minutes before moving on. Residual eDNA indicates they were there but is that a measure of local diversity? The authors need to show more knowledge about the species that they are assessing and the habitat use.
13. L591. “…mixed or homogenized through movement…” add water and species before movement
14. L538. “…beach seining is ineffective outside shallow soft sediments…” rather a bold unsubstantiated statement. Ineffective at what?

Reviewer 3 ·

Basic reporting

I could not find these data in the paper or in the supplemental files either, I feel it is very important that there be a table listing the collecting data all of the collecting events both eDNA and capture, data such as gps coordinates, depth, date and time of collection and any other information you have should be listed here. I also wish the map was presented earlier in the paper.

Experimental design

The experimental design is reasonable but it is problematic that the capture vs eDNA sampling events took place such a large amount of time apart from each other making comparison difficult, I do understand it is not always possible to have perfect sampling but I think more attention should be given to the fact the capture and eDNA sampling did not happen at the same time which limits your ability to truly compare and contrast the two sampling methods. Because I couldn't easy see these sampling dates and locations as mentioned above I don't know how large of factor this is or isn't based off your description in the paper.

Validity of the findings

These findings are reasonable to extend to the fish taxa that you have in your DNA barcode reference database but the title and phrasing throughout the text of 'marine biodiversity' is problematic there is so much to marine biodiversity than fishes, e.g. invertebrates, bacteria, viruses etc I feel strongly that these findings can only be applied to the fish taxa you were targeting and not all of 'marine biodiversity'.

Additional comments

Overall this paper is a well, written and executed example of comparing eDNA metabarcoding results to traditional capture based survey methods. These studies showing how eDNA compares to traditional sampling methods are badly needed and it is important that as many of these types of studies in as many different systems as possible are completed to help us understand how to compare the results of these different methods. I feel strongly that studies like this one should be prioritized and supported by the community. Even know this study dose not have a perfect experimental design it is still useful information and should be published but I would also like to reiterate that making sweeping statements about marine biodiversity based on a fish study is not appropriate and would want to see a change to the title and to the wording in the text that expressly limits these findings to the fish taxa being targeted.

---

## Round 0.2 · accepted · Accept

Thank you for the revision of the manuscript. I hereby certify that you have adequately taken into account the reviewer's comments and improved the manuscript accordingly. Based on my assessment as an Academic Editor, your manuscript is now ready for publication.

Reviewer 2 ·

Basic reporting

Good structure and sentencing

Experimental design

Adequate testing of stated hypotheses

Validity of the findings

Conclusions are justified with the data and analyses presented

Additional comments

The authors have, in my opinion, strengthened their paper by considering the comments of the 3 reviewers. Hence, the MS should be accepted for publication.